# Feature Importance of Acute Rejection among Black Kidney Transplant Recipients by Utilizing Random Forest Analysis: An Analysis of the UNOS Database

**DOI:** 10.3390/medicines8110066

**Published:** 2021-11-02

**Authors:** Charat Thongprayoon, Caroline C. Jadlowiec, Napat Leeaphorn, Jackrapong Bruminhent, Prakrati C. Acharya, Chirag Acharya, Pattharawin Pattharanitima, Wisit Kaewput, Boonphiphop Boonpheng, Wisit Cheungpasitporn

**Affiliations:** 1Division of Nephrology and Hypertension, Department of Medicine, Mayo Clinic, Rochester, MN 55905, USA; 2Division of Transplant Surgery, Mayo Clinic, Phoenix, AZ 85054, USA; Jadlowiec.Caroline@mayo.edu; 3Renal Transplant Program, University of Missouri-Kansas City School of Medicine/Saint Luke’s Health System, Kansas City, MO 64131, USA; napat.leeaphorn@gmail.com; 4Excellence Center for Organ Transplantation, Faculty of Medicine, Ramathibodi Hospital, Mahidol University, Bangkok 10400, Thailand, Division of Infectious Diseases, Department of Medicine, Faculty of Medicine, Ramathibodi Hospital, Mahidol University, Bangkok 10400, Thailand; jbruminhent@gmail.com; 5Division of Nephrology, Texas Tech Health Sciences Center El Paso, El Paso, TX 79905, USA; prakrati.c.acharya@gmail.com (P.C.A.); drchiragacharya1985@gmail.com (C.A.); 6Department of Internal Medicine, Faculty of Medicine, Thammasat University, Pathum Thani 12120, Thailand; 7Department of Military and Community Medicine, Phramongkutklao College of Medicine, Bangkok 10400, Thailand; 8Division of Nephrology, University of Washington, Seattle, WA 98195, USA; boonpipop.b@gmail.com

**Keywords:** black, race, kidney transplant, transplantation, risk factors, feature importance, machine learning, artificial intelligence, nephrology, precision medicine, personalized medicine, individualized medicine

## Abstract

**Background**: Black kidney transplant recipients have worse allograft outcomes compared to White recipients. The feature importance and feature interaction network analysis framework of machine learning random forest (RF) analysis may provide an understanding of RF structures to design strategies to prevent acute rejection among Black recipients. **Methods:** We conducted tree-based RF feature importance of Black kidney transplant recipients in United States from 2015 to 2019 in the UNOS database using the number of nodes, accuracy decrease, gini decrease, times_a_root, *p* value, and mean minimal depth. Feature interaction analysis was also performed to evaluate the most frequent occurrences in the RF classification run between correlated and uncorrelated pairs. **Results:** A total of 22,687 Black kidney transplant recipients were eligible for analysis. Of these, 1330 (6%) had acute rejection within 1 year after kidney transplant. Important variables in the RF models for acute rejection among Black kidney transplant recipients included recipient age, ESKD etiology, PRA, cold ischemia time, donor age, HLA DR mismatch, BMI, serum albumin, degree of HLA mismatch, education level, and dialysis duration. The three most frequent interactions consisted of two numerical variables, including recipient age:donor age, recipient age:serum albumin, and recipient age:BMI, respectively. **Conclusions:** The application of tree-based RF feature importance and feature interaction network analysis framework identified recipient age, ESKD etiology, PRA, cold ischemia time, donor age, HLA DR mismatch, BMI, serum albumin, degree of HLA mismatch, education level, and dialysis duration as important variables in the RF models for acute rejection among Black kidney transplant recipients in the United States.

## 1. Introduction

Black kidney transplant recipients have worse allograft outcomes compared to White recipients [1,2,3,4,5,6,7]. Previously identified factors possibly responsible for these disparities have include lower living donor rates [5], longer dialysis vintage [8], longer cold ischemia times [9], delays in listing and time on transplant waiting list [10,11], greater rates of marginal donor kidneys [9,12,13], higher rates of delayed graft function (DGF) [14], less favorable human leucocyte antigen (HLA) matching [9,15,16,17], variability in the pharmacokinetics of immunosuppressive drugs and immunologic responsiveness [18,19], and more comorbidities, such as hypertension [20] and diabetes mellitus [21]. Such discrepancies have been thought to contribute to a higher rate of acute rejection in Black recipients, resulting in increased allograft loss [14,22,23,24,25]. Despite advancements in modern-era immunosuppressive agents, the beneficial effects of immunosuppression specific to acute rejection remain less apparent in Black recipients compared to White recipients [7,22,26,27]. Furthermore, acute rejection in Black kidney transplant recipients is more likely to be steroid-resistant [28].

Recent investigations have demonstrated that machine learning approaches are superior to traditional statistical methods in various clinical scenarios [29,30]. Random forest (RF) is a widely used machine learning approach that effectively predicts outcomes [31] by utilizing a combination of tree predictors [32]. The RF algorithm randomly generates bootstrapped datasets that can be used to train an ensemble of decision trees, which determine an outcome by a majority “vote” [32]. As a type of robust nonparametric model, RF can simulate complex relationships and does not depend on the data distribution as is the case in logistic regression [31]. Whereas most traditional statistical approaches, such as linear regression and logistic regression, indicate which variables are significant with measures such as *p*-value and *t*-statistics, variable importance by RF is determined by how much each variable decreases the node impurity (gini decrease), number of nodes, accuracy, mean minimal depth, and times_a_root (total number of trees in which X_j_ is used for splitting the root node) [31,33]. Recently, RF has increasingly been applied to medicine, including solid organ transplantation [34,35,36], and there is great potential to use the RF approach to improve outcomes among Black kidney transplant recipients. Furthermore, the feature interaction network analysis framework of RF may also provide an understanding of the interaction among multiple features in order to design strategies to prevent acute rejection and explore the mechanisms of variable interactions influencing acute rejection among Black kidney transplant recipients [37].

In this study of the UNOS/OPTN database from 2015 through 2019, we aimed to assess the risk factors and feature importance of acute rejection among Black kidney transplant recipients by utilizing RF vs. traditional multivariable logistic regression analysis.

## 2. Materials and Methods

### 2.1. Data Source and Study Population

The Organ Procurement and Transplantation Network (OPTN)/United Network for Organ Sharing database (UNOS) database was used for analysis of this study. The OPTN/UNOS database contains patient-level data of all transplant events in the United States. All adult (age ≥18 years) end-stage kidney disease patients who received kidney-only kidney transplants from 2015 to 2019 were screened. Only Black patients were included in this study. If patients had multiple kidney transplants during the study period, the first kidney transplant was selected for analysis. This study was approved by the Mayo Clinic Institutional Review Board (#21-007698) and the UNOS/OPTN data is publicly available and de-identified.

### 2.2. Data Collection

Comprehensive recipient-, donor-, and transplant-related variables in the OPTN/UNOS database were extracted. All the extracted variables had less than 10% of missing data. Missing data was imputed through multivariable imputation by the chained equation (MICE) method [38].

The primary outcome was acute rejection reported by transplant centers to the OPTN/UNOS within 1 year after kidney transplant. The UPTN/UNOS database did not specify the date of acute rejection occurrence.

### 2.3. Machine Learning Variable Importance Analysis

Variable importance was performed using the “randomForest” package and interpreted and visualized by the “randomForestExplainer” [39] packages in R 4.0.2. Random forests are ensemble classifiers that aggregate the results of many individual decision trees. We used the ‘randomForest’ R package [40] with two hyperparameters: the number of training trees (nTree) and the number of predictors to consider at each split point (mTry). The default settings of nTree = 500 and mTry as the square root of the number of predictor variables were used in this study. To avoid the bias of analysis of variable importance, various indicators (number of nodes, accuracy decrease, Gini decrease, times_a_root (total number of trees in which X_j_ is used for splitting the root node), *p* value, and mean minimal depth) were selected to represent different perspectives and to comprehensively evaluate the importance of features [31,33]. The Gini impurity measures the frequency at which any element of the dataset will be mislabeled when it is randomly labeled. The minimum value of the Gini Index is 0. This happens when the node is pure, indicating that all the contained elements in the node are of one unique class. “Gini_decrease” indicates the decrease in the Gini impurity index, and “accuracy_decrease” refers to the mean decrease of prediction accuracy after the corresponding predictor was permuted [39].

The importance of each variable can be expressed using other metrics, such as mean minimal depth, times_a_root, accuracy decrease, and Gini decrease.

### 2.4. Statistical Analysis

Continuous variables were presented as mean ± standard deviation (SD) for normally distributed data, or median with interquartile range (IQR) for non-normally distributed data. Categorical variables were presented as number with percentage. The difference in clinical characteristics between patients with and without rejection were tested using the student’s t-test or Wilcoxon’s rank sum test as appropriate for continuous variables, and Chi-squared test for categorical variables. For traditional analysis to identify independent predictors for rejection, backward stepwise multivariable logistic regression with inclusion of variables whose *p*-value in univariable analysis <0.05 was performed.

All analyses were performed using R, version 4.0.3 (RStudio, Inc., Boston, MA, USA; http://www.rstudio.com/ (accessed on 15 January 2021)). We used the “randomForest” and “randomForestExplainer” packages in R 4.0.2. for machine learning analysis [39], and the MICE command in R for multivariable imputation by chained equation [38].

## 3. Results

A total of 22,687 black kidney transplant recipients were eligible for analysis. Of these, 1330 (6%) had acute rejection within 1 year after kidney transplant. Table 1 compared the recipient-, donor-, and transplant-related characteristics between patients with and without rejection. Patients with rejection were younger, more likely to have a glomerular kidney disease etiology, have longer dialysis duration, and be HIV-seropositive. They were less likely to be diabetic or receive a living donor kidney transplant and more likely to have delayed graft function. They were also more likely to be kidney retransplants, have a higher PRA, have a higher total number of HLA mismatches, and carry public insurance. With regard to immunosuppression, patients with rejection were less likely to receive depleting induction (e.g., thymoglobulin, alemtuzumab) and were more likely to receive basiliximab. They were also more likely to be on cyclosporine, mycophenolate, azathioprine, and mTOR inhibitors for maintenance immunosuppression.

### 3.1. Traditional Analysis

The multi-collinearity of continuous variables was assessed by a correlation matrix, which demonstrated no significant multi-collinearity (Figure 1).

Using traditional analysis with backward stepwise multivariable logistic regression (Table 2), the independent predictors for increased acute rejection risk included kidney retransplantion, dialysis duration ≥1 years, a PRA of 81–100, HIV infection, ECD deceased donor utilization, a higher total number of HLA mismatches, delayed graft function, basiliximab induction, and the use of cyclosporine, azathioprine, and mTOR inhibitors for maintenance immunosuppression. In contrast, the independent predictors for decreased rejection risk included older recipient age and the use of thymoglobulin and alemtuzumab for induction.

### 3.2. Machine Learning Variable Importance

#### 3.2.1. Distribution of Minimal Depth

Figure 2 demonstrates variables based on the minimum depth locations between the tree and the number of trees. The minimal depth for a variable in a tree is equal to the depth of the node which splits on that variable and is the closest to the root of the tree. If it is low, then a number of observations are divided into groups on the basis of this variable. From the top 10 variables with the smallest mean value of minimal depth plotted in Figure 2, recipient age, cold ischemia time, BMI, PRA, and serum albumin are the top 5 variables used to split trees at the root. The RF model built 500 trees with no limit to the maximum number of terminal nodes in a tree. It is evident that trees were split until a depth of 11.

#### 3.2.2. Importance Measures

Table 3 demonstrates the measures of importance (mean minimal depth, number of nodes, accuracy decrease, gini decrease, number of trees, times_a_root, and *p* value) for all variables in the forest. The following variables are the top five variables based on different important measures:Mean minimal depth: recipient age, cold ischemia time, BMI, PRA, and serum albumin are the top five variables used to split trees at the root.Number of nodes: cold ischemia time, BMI, donor age, recipient age, and serum albumin.Decrease in accuracy: cold ischemia time, age, donor age, KDPI group, and number of transplants.Decrease in Gini: BMI, cold ischemia time, recipient age, donor age, and serum albumin.Number of trees: BMI, cold ischemia time, recipient age, donor age, and serum albumin.Times_a_root: recipient age, retransplant, cause of ESKD, DGF, and basiliximab induction.

#### 3.2.3. Multi-way Importance Plot

The multi-way importance plot reveals the relation between the 3 measures of importance and labels 10 variables that scored best when it comes to these 3 measures.

The first multi-way importance plot (Figure 3) centers on three important measures acquired from the structure of trees in the forest, including (1) mean depth of the first split on the variable, (2) number of trees in which the root is split on the variable, and (3) the total number of nodes in the forest that split on that variable. These top 10 relative variables of importance in the RF models for acute rejection are based on the minimum average depth and the number of nodes, and the times to root include age, cause of ESKD, PRA, education level, cold ischemia time, HLA DR mismatch, BMI, serum albumin, degree of HLA mismatch, and donor age.

The second multi-way importance plot (Figure 4) reveals the important measures that emerge from the role of the variables in the prediction of acute rejection, including a decrease in accuracy and a decrease in Gini, with additional information on the *p*-value based binomial distribution of the number of nodes split on the variable implying that the variables are randomly drawn to form splits. After combining the mean decrease in Gini, decrease in accuracy, and *p* values of these features, the top variables for acute rejection include cold ischemia time, recipient age, donor age, BMI, serum albumin, PRA, degree of HLA mismatch, causes of ESKD, dialysis duration, and HLA-DR mismatch.

#### 3.2.4. Compare Rankings of Variables

Figure 5 exhibits the bilateral relations between the rankings of variables according to the selected importance measures. It demonstrates that the RF parameters of importance are ascertained to have correlations among each other, thereby implying the reliability of each of these parameters to rank the variable importance. The top correlations among themeasures of importance include decrease in gini:number of nodes, decrease in gini:mean minimal depth, and mean minimal depth: number of nodes.

#### 3.2.5. Variable Interactions

Feature interactions with the most frequent occurrences in the RF classification run between correlated and uncorrelated pairs. Figure 6 outlines the 30 top interactions of the variables according to the mean of conditional minimal depth, a generalization of minimal depth that measures the depth of the second variable in a tree of which the first variable is a root (a subtree of a tree from the forest).

To be comparable to the normal minimal depth, 1 is subtracted so that 0 is the minimum. Smaller values of the mean conditional depth with associated higher unconditional depth, as well as increased occurrences, indicate interaction effects (Table 4). The interactions considered are those with the following variables as first (root variables): cold ischemia time, recipient age, donor age, BMI, PRA, cause of ESKD, serum albumin, total number of HLA mismatches, dialysis duration, HLA-DR mismatches, allocation type, education level, CMV status, HLA-B mismatch, HLA-A mismatch, and all plausible values of the second variable. The three most frequent interactions consist of two variables, including recipient age:donor age, recipient age:serum albumin, and recipient age:BMI, respectively.

## 4. Discussion

In this study, using tree-based RF feature importance and the feature interaction network analysis framework, we were able to demonstrate important variables in the RF models for acute rejection among Black kidney transplant recipients using the number of nodes, accuracy decrease, gini decrease, times_a_root, *p* value, and mean minimal depth. These identified risk factors for rejection included recipient age, cause of ESKD, PRA, cold ischemia time, donor age, HLA DR mismatch, BMI, serum albumin, degree of HLA mismatch, education level, and dialysis duration.

By comparison, traditional multivariable logistic regression analysis showed that younger recipient age; kidney retransplantation; ECD deceased donor utilization; dialysis duration; PRA; recipient HIV seropositivity; degree of HLA mismatch; DGF; basiliximab induction; and cyclosporine, azathioprine, and mTOR inhibitor-based immunosuppression were independent risk factors for acute rejection among Black kidney transplant recipients. Whereas some important variables from traditional logistic regression analysis are not listed as important variables in the RF approach, these factors are still variables that were used to predict acute rejection outcomes among Black kidney transplant recipients (as shown in Figure 7).

RF is an ensemble of decision trees. Many trees produced in a particular “random” way build a RF [41]. Each tree is constructed from a diverse sample of rows, and at individual nodes, a different sample of features is chosen for splitting. Each of the trees produces its own individual prediction. These predictions are subsequently averaged to generate a single result. Averaging strengthens a RF to be better than a single decision tree and thereby increases its accuracy and lessens overfitting. The average of these models evens out the variance, resulting in an error reduction that is both low in bias and low in variance. This nonparametric and nonlinear machine learning RF method can resist noise and is expected to build accurate prediction models using aggregated data. In addition, RF works well on large datasets, especially when there are many categorical independent variables and unbalanced data [41], as in our OPTN/UNOS dataset. Conversely, a logistic regression analysis approach, which uses a generalized linear equation and the stepwise variable selection method, is based on the likelihood ratio test to describe the directed dependencies among a set of variables. To do so, a number of statistical assumptions must be met. Common concerns include overfitting (rule of 10) as well as outliers. As a result, logistic regression inherently has bias and low variance due to the rigid nature of the shape of the line.

Previous studies have demonstrated higher PRA, longer cold ischemia time, increased HLA mismatches, HLA DR mismatches, and longer dialysis duration as important generalizable risk factors for acute rejection among kidney transplant recipients. From our current study using RF, we demonstrate that some of these established variables, such as cold ischemia time and HLA DR mismatch, are not listed as independent predictors for acute rejection among Black kidney transplant recipients in traditional logistic regression analysis. Furthermore, we also found that BMI, cause of ESKD, serum albumin, donor age, and education level are important variables in RF, but not in traditional analysis. Given that the RF algorithm has increasingly been applied in medicine and transplantation [34,35,36], it is important to recognize these unique RF model variables for acute rejection among Black kidney transplant recipients. A good prediction model begins with a great feature selection process. Understanding these variables in our study using a national database will help each transplant center to develop their individualized RF model, prognosticate the risk of rejection among Black kidney transplant recipients, and develop strategies to prevent these serious events.

In addition to the identification of feature importance for acute rejection in Black kidney transplant recipients in the RF model, we also conducted feature interaction network analysis. A great benefit of the tree structure is the understanding of the interaction between variables. For example, if the split in a parent is by one variable and by another variable in the daughter node, it can be concluded that there is an interaction between these two variables [14,24,25]. Interactions also become apparent as common occurrences of variable combinations. Thus, both the pair frequency and the associated distances are informative with regard to the interaction effects. From the findings of our study, the three most frequent interactions consist of two numerical variables, including recipient age:donor age, recipient age:serum albumin, and recipient age:BMI, respectively. These findings from the feature interaction network analysis may help to determine the important effect modifiers of acute rejection risk among Black kidney transplant recipients.

Our study has several limitations. Given the nature of the UNOS database, we do not have details on the factors leading to acute rejection such as immunosuppression levels, medication non-adherence, donor-specific antibodies, or infection prior to episodes of acute rejection. Thus, we aimed to investigate RF feature importance with feature interaction network analysis as an initial step to create an RF prediction model. Additional investigations are needed to incorporate the findings of our study into other important variables such as crossmatch results, the presence of DSA, and follow-up data to construct a RF prediction model with a high predictive performance for acute rejection among Black kidney transplant recipients. In addition, future studies assessing tree-based RF feature importance and the feature interaction network analysis framework for acute rejection among the general kidney transplant recipient populations are needed.

## 5. Conclusions

In conclusion, the application of tree-based RF feature importance and the feature interaction network analysis framework identified the recipient age, ESKD etiology, PRA, cold ischemia time, donor age, HLA DR mismatch, BMI, serum albumin, degree of HLA mismatch, education level, and dialysis duration as important variables in the RF models for acute rejection among Black kidney transplant recipients in the United States.

## Figures and Tables

**Figure 1 medicines-08-00066-f001:**
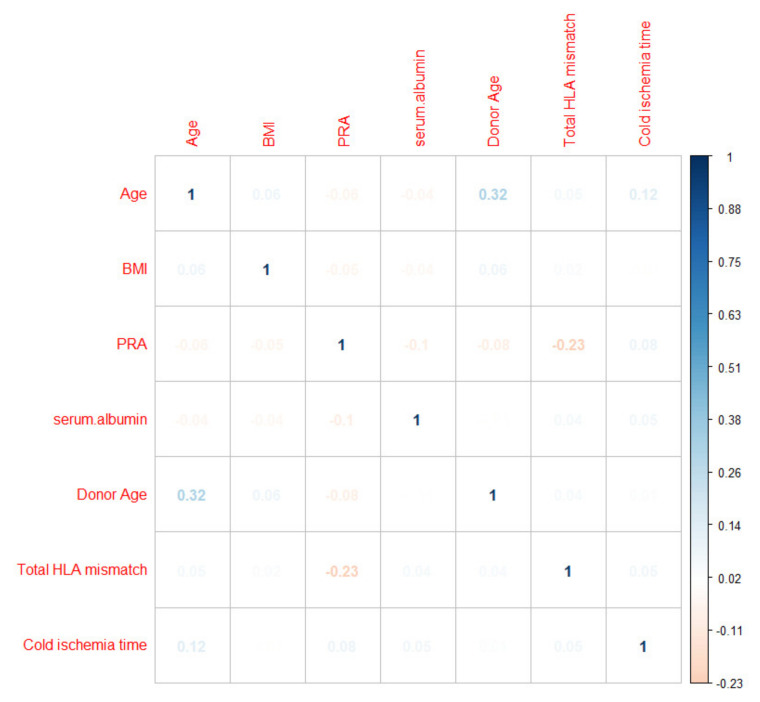
Correlation matrix assessing the pair-wise correlation between the continuous variables.

**Figure 2 medicines-08-00066-f002:**
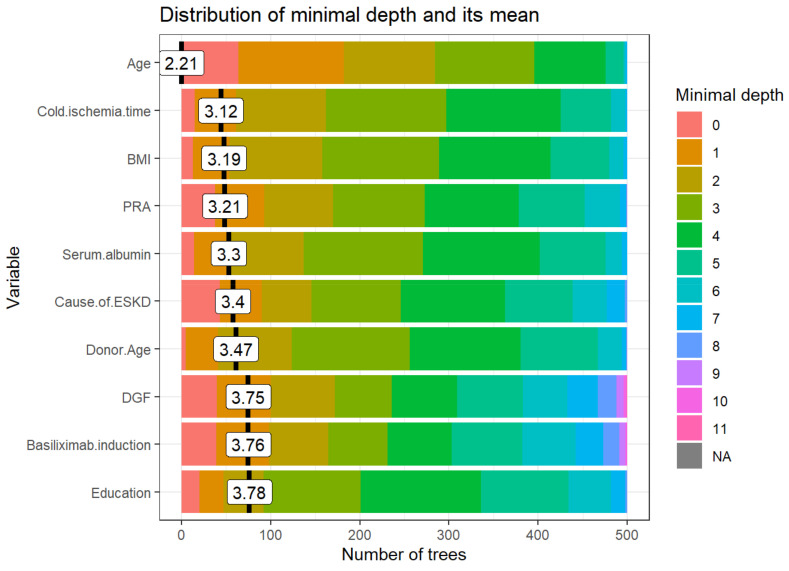
The distribution of the minimal depth among the trees of the forest for the important variables is presented in distinctive colors for the individual level of minimal depth. The mean of the distribution is demonstrated by a vertical black bar with a value label on it, and the scale of the X-axis ranges from 0 to the maximum number of trees in which any variable was utilized for splitting. Abbreviations: BMI- Body mass index; DGF—Delayed graft function; ESKD—End-stage kidney disease; HLA—Human leukocyte antigen; PRA—Panel reactive antibody.

**Figure 3 medicines-08-00066-f003:**
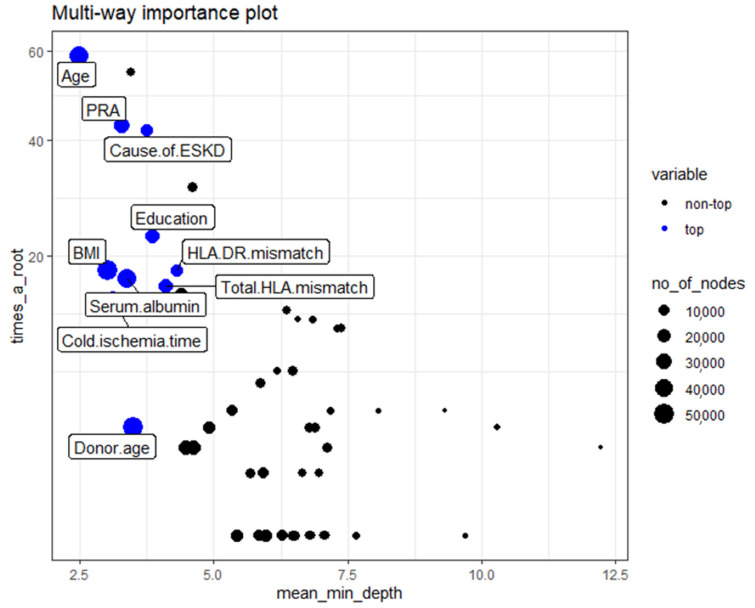
Classification of the top and not top variables based on the minimum average depth, number of nodes, and times to root. The 10 top variables are highlighted in blue. The size of points reflects the number of nodes split on the variable. Times_a_root: total number of trees in which the variable is used for splitting the root node. Mean_minimal_depth: mean minimal depth. Abbreviations: BMI—Body mass index; ESKD—Endstage kidney disease; HLA—Human Leukocyte antigen; PRA—Panel reactive antibody.

**Figure 4 medicines-08-00066-f004:**
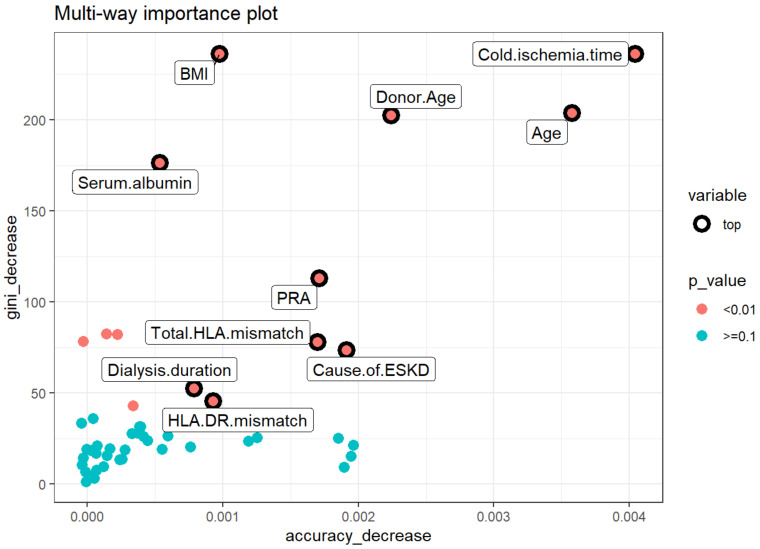
Multiway feature importance analysis of the acute rejection combining the mean decrease in Gini, decrease in accuracy, and *p* values of the features (pink circle; *p* < 0.01). Abbreviations: BMI—Body mass index; ESKD—End-stage kidney disease; HLA—Human leukocyte antigen; PRA—Panel reactive antibody.

**Figure 5 medicines-08-00066-f005:**
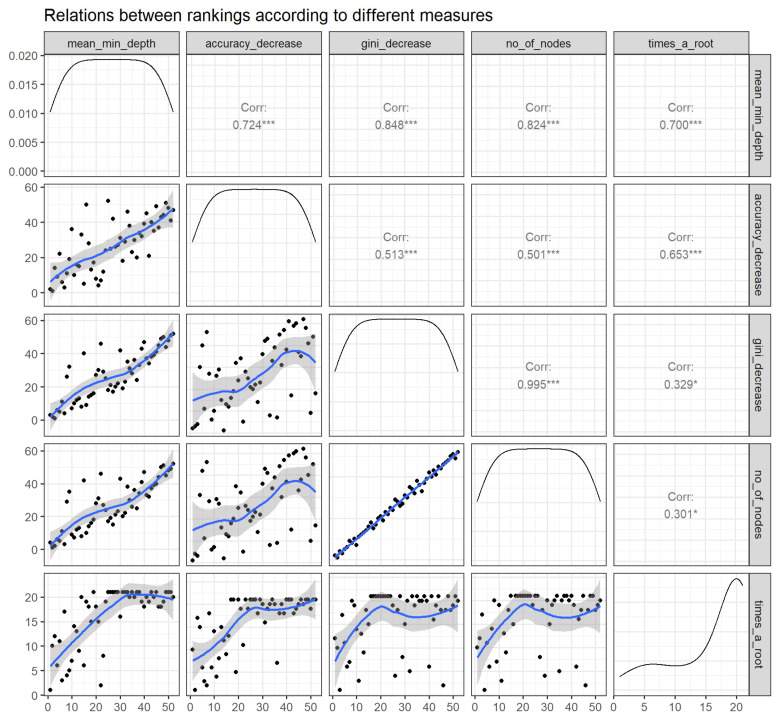
Relations between rankings according to different measures (panels in the lower triangle of the grid displaying distribution of rankings of all predictor variables with black dots accompanying a blue trend line) as well as correlation coefficient over rankings of any 2 parameters (panels in the upper triangle of the grid). Abbreviations: No—Number; min—Minimum; Corr—correlation. * *p* < 0.05, *** *p* < 0.001.

**Figure 6 medicines-08-00066-f006:**
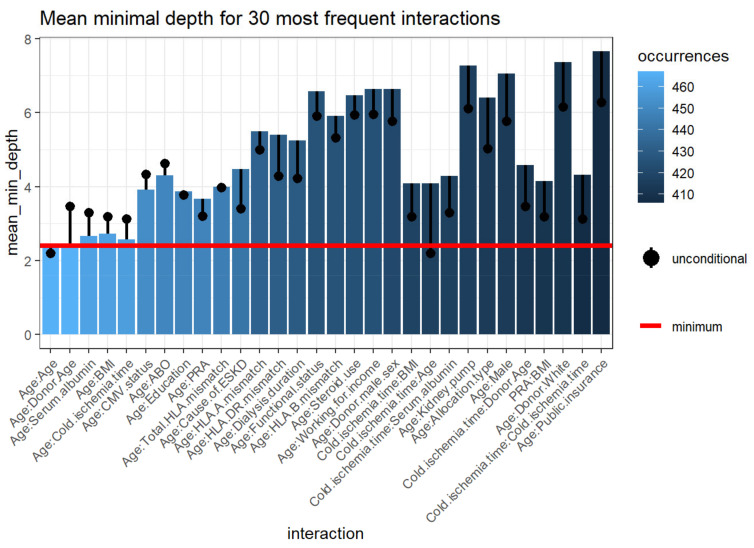
Mean minimum depth for the 30 most prevalent interactions variables. The bar displaying the mean conditional minimal depth and the line the unconditional mean minimal depth. The horizontal line presents the minimal value of the depicted statistic among interactions for which it was calculated. Abbreviations: BMI—Body mass index; CMV—Cytomegalovirus; ESKD—End-stage kidney disease; HLA—Human leukocyte antigen; PRA—Panel reactive antibody.

**Figure 7 medicines-08-00066-f007:**
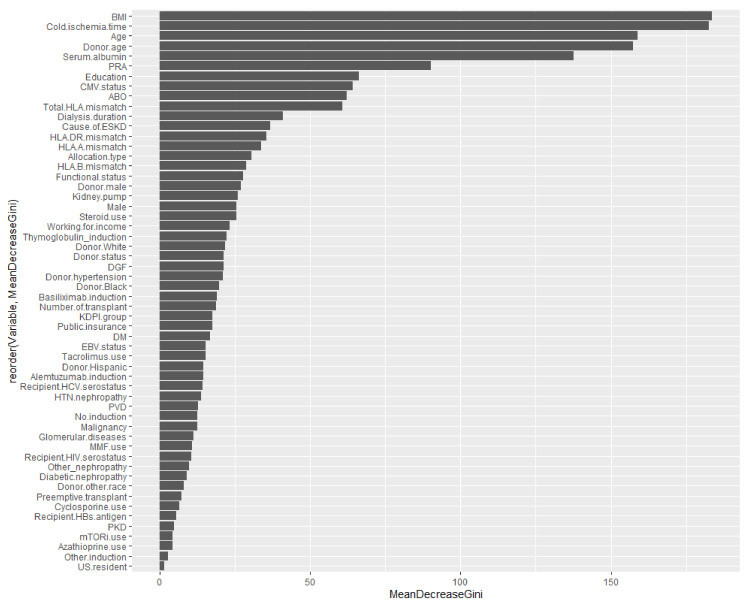
Variable importance summarized from Table 1 (The order of these listed variables based on mean decrease in Gini). Abbreviation: BMI—Body mass index; CMV—Cytomegalovirus; DGF—Delayed graft function; DM—Diabetes mellitus; EBV—Epstein-Barr virus; ESKD—End-stage kidney disease; HBs—Hepatitis B surface; HCV—Hepatitis C virus; HIV—Human immunodeficiency virus; HTN—Hypertension; KDPI—Kidney donor profile index; MMF—Mycophenolate, mTORi—Mammalian target of rapamycin inhibitor; PKD—Polycystic kidney disease; PRA—Panel reactive antibody; PVD—Peripheral vascular disease.

**Table 1 medicines-08-00066-t001:** Clinical characteristics between kidney transplant recipients with and without rejection.

	All	Rejection	No Rejection	*p*-Value
	(*n* = 22,687)	(*n* = 1330)	(*n* = 21,357)	
Recipient Age (year)	51.4 ± 12.6	47.8 ± 13.2	51.6 ± 12.6	<0.001
	52 (43–61)	48 (37–58)	53 (43–61)	
Recipient male sex	13,635 (60)	817 (61)	12,818 (60)	0.31
ABO blood group				0.34
- A	6452 (28)	368 (28)	6084 (28)	
- B	4334 (19)	239 (18)	4095 (19)	
- AB	1255 (6)	68 (5)	1187 (6)	
- O	10,646 (50)	655 (49)	9991 (47)	
Body mass index (kg/m^2^)	29.3 ± 5.7	29.2 ± 5.5	29.3 ± 5.7	0.33
	29.0 (25.1–33.4)	28.9 (25.1–33.2)	29.0 (25.1–33.4)	
Kidney retransplant	2413 (11)	222 (17)	2191 (10)	<0.001
Dialysis duration				<0.001
- Preemptive or <1 year	3585 (16)	159 (12)	3426 (16)	
- 1–3 years	4069 (18)	245 (18)	3824 (18)	
- >3 years	15,033 (66)	926 (70)	14,107 (66)	
Cause of end-stage kidney disease				<0.001
- Diabetes mellitus	6460 (28)	313 (24)	6147 (29)	
- Hypertension	8189 (36)	455 (34)	7734 (36)	
- Glomerular disease	4027 (18)	295 (22)	3732 (17)	
- PKD	839 (4)	36 (3)	803 (4)	
- Other	3172 (14)	231 (17)	2941 (14)	
Comorbidity				
- Diabetes mellitus	8253 (36)	417 (31)	7836 (37)	<0.001
- Malignancy	1580 (7)	85 (6)	1495 (7)	0.4
- Peripheral vascular disease	2119 (9)	109 (8)	2010 (9)	0.14
PRA	0 (0–48)	0 (0–69)	0 (0–47)	<0.001
Positive HCV serostatus	1825 (8)	118 (9)	1707 (8)	0.25
Positive HBs antigen	340 (2)	18 (1)	322 (2)	0.65
Positive HIV serostatus	767 (3)	67 (5)	700 (3)	0.001
Functional status				0.84
- 10–30%	50 (0)	2 (0.2)	48 (0.2)	
- 40–70%	11,869 (52)	700 (53)	11,169 (52)	
- 80–100%	10,768 (47)	628 (47)	10,140 (47)	
Working income	5883 (26)	320 (24)	5563 (26)	0.11
Public insurance	18,504 (82)	1117 (84)	17,387 (81)	0.02
US resident	22,597 (99)	1327 (99)	21,270 (99)	0.31
Undergraduate education or above	12,405 (55)	708 (53)	11,697 (55)	0.28
Serum albumin	4.0 ± 0.6	3.9 ± 0.5	4.0 ± 0.6	0.32
	4.0 (3.6–4.3)	4.0 (3.6–4.3)	4.0 (3.6–4.3)	
Kidney donor status				0.047
- Non-ECD deceased	17,052 (75)	1030 (77)	16,022 (75)	
- ECD deceased	2482 (11)	145 (11)	2337 (11)	
- Living	3153 (14)	155 (12)	2998 (14)	
Donor age	38.4 ± 4.8	38.2 ± 15.0	38.2 ± 14.8	0.99
	38 (27–50)	39 (27–50)	38 (27–50)	
Donor male sex	13,064 (58)	796 (60)	12,268 (57)	0.08
Donor race				0.57
- White	13,784 (61)	817 (61)	12,967 (61)	
- African American	5918 (26)	328 (25)	5590 (26)	
- Hispanic	2266 (10)	143 (11)	2123 (10)	
- Other	719 (3)	42 (3)	677 (3)	
History of hypertension in donor	5477 (24)	310 (23)	5167 (24)	0.46
KDPI				0.02
- Living donor	3153 (14)	155 (12)	2998 (14)	
- KDPI < 85	17,892 (79)	1093 (82)	16,889 (79)	
- KDPI ≥ 85	1552 (7)	82 (6)	1470 (7)	
HLA mismatch				
- A	2 (1–2)	2 (1–2)	2 (1–2)	0.64
- B	2 (1–2)	2 (1–2)	2 (1–2)	0.48
- DR	1 (1–2)	1 (1–2)	1 (1–2)	<0.001
- ABDR	5 (4–5)	5 (4–5)	5 (4–5)	0.008
Cold ischemia time	15.8 ± 9.8	16.1 ± 9.5	15.8 ± 9.8	0.41
	15.5 (9.1–21.7)	15.2 (9.8–21.4)	15.5 (9.0–21.7)	
Allocation type				0.03
- Local	16,718 (74)	973 (73)	15,745 (74)	
- Regional	2821 (12)	142 (11)	2679 (13)	
- National	3147 (14)	215 (16)	2633 (14)	
Kidney on pump	9496 (42)	557 (42)	8939 (42)	0.99
Delay graft function	6720 (30)	494 (37)	6226 (29)	<0.001
EBV status				0.54
- Low risk	122 (1)	5 (0.4)	117 (0.5)	
- Moderate risk	21,200 (93)	1239 (93)	19,961 (93)	
- High risk	1365 (6)	86 (6)	1279 (6)	
CMV status				0.55
- D-/R-	2531 (11)	149 (11)	2382 (11)	
- D-/R+	6554 (29)	374 (28)	6180 (29)	
- D+/R+	10,398 (46)	632 (48)	9766 (46)	
- D+/R-	3204 (14)	175 (13)	3029 (14)	
Induction immunosuppression				
- Thymoglobulin	14,376 (63)	803 (60)	13,573 (64)	0.02
- Alemtuzumab	3792 (18)	197 (15)	3595 (17)	0.05
- Basiliximab	3684 (16)	289 (22)	3395 (16)	<0.001
- Other	328 (1)	9 (0.7)	319 (1)	0.02
- No induction	1547 (7)	96 (7)	1451 (7)	0.55
Maintenance Immunosuppression				
- Tacrolimus	20,689 (91)	1203 (90)	19,486 (91)	0.33
- Cyclosporine	184 (1)	19 (1)	165 (1)	0.01
- Mycophenolate	20,907 (92)	1250 (94)	19,657 (92)	0.01
- Azathioprine	65 (0.3)	8 (0.6)	57 (0.3)	0.03
- mTOR inhibitors	62 (0.3)	8 (0.6)	54 (0.3)	0.02
- Steroid	16,131 (71)	954 (72)	15,177 (71)	0.60

Abbreviations: BMI—Body mass index; CMV—Cytomegalovirus; DGF—Delayed graft function; DM—Diabetes mellitus; EBV—Epstein-Barr Virus; ESKD—End-stage kidney disease; HBs—Hepatitis B surface; HCV—Hepatitis C virus; HIV—Human immunodeficiency virus; HLA—Human leukocyte antigen; KDPI—Kidney donor profile index; mTORi—Mammalian target of rapamycin inhibitors; PRA—Panel reactive antibody; PVD—Peripheral vascular disease.

**Table 2 medicines-08-00066-t002:** Univariable and multivariable logistic regression analysis for acute rejection.

Variable	Univariate Analysis		Multivariate Analysis	
	OR (95% CI)	*p*-Value	OR (95% CI)	*p*-Value
Recipient Age (per 5-year increase)	0.89 (0.87–0.91)	<0.001	0.88 (0.86–0.90)	<0.001
Recipient male sex	1.06 (0.95–1.19)	0.31		
ABO blood group				
- A	0.92 (0.81–1.05)	0.23		
- B	0.89 (0.76–1.04)	0.13		
- AB	0.87 (0.68–1.13)	0.30		
- O	1 (ref)	-		
Body mass index (kg/m^2^)	0.98 (0.93–1.03)	0.34		
Kidney retransplant	1.75 (1.51–2.04)	<0.001	1.50 (1.24–1.81)	<0.001
Dialysis duration				
- Preemptive or <1 year	1 (ref)	-	1 (ref)	-
- 1–3 years	1.38 (1.12–1.69)	0.002	1.28 (1.04–1.58)	0.02
- >3 years	1.41 (1.19–1.68)	<0.001	1.26 (1.04–1.51)	0.02
Cause of end-stage kidney disease				
- Diabetes mellitus	0.87 (0.75–1.00)	0.055		
- Hypertension	1 (ref)			
- Glomerular disease	1.34 (1.15–1.56)	<0.001		
- PKD	0.76 (0.54–1.08)	0.11		
- Other	1.34 (1.13–1.57)	<0.001		
Comorbidity				
- Diabetes mellitus	0.79 (0.70–0.89)	<0.001		
- Malignancy	0.91 (0.72–1.13)	0.39		
- Peripheral vascular disease	0.86 (0.70–1.05)	0.13		
- PRA				
- 0	1 (ref)	-	1 (ref)	-
- 1–20	1.18 (0.98–1.42)	0.09	1.18 (0.98–1.43)	0.08
- 21–80	0.96 (0.81–1.12)	0.59	0.97 (0.82–1.14)	0.69
- 81–100	1.51 (1.31–1.74)	<0.001	1.39 (1.16–1.67)	<0.001
Positive HCV serostatus	1.12 (0.92–1.36)	0.25		
Positive HBs antigen	0.90 (0.56–1.45)	0.65		
Positive HIV serostatus	1.57 (1.21–2.02)	0.001	1.35 (1.03–1.76)	0.03
Functional status <80%	1.01 (0.90–1.13)	0.85		
Working income	0.90 (0.79–1.02)	0.11		
Public insurance	1.20 (1.03–1.39)	0.02		
US resident	1.81 (0.57–5.73)	0.31		
Undergraduate education or above	0.94 (0.84–1.05)	0.28		
Serum albumin (per 1-g/dL increase)	0.95 (0.86–1.05)	0.33		
Kidney donor status				
- Non-ECD deceased	1 (ref)	-	1 (ref)	-
- ECD deceased	0.97 (0.81–1.15)	0.70	1.23 (1.02–1.48)	0.03
- Living	0.80 (0.68–0.96)	0.01	0.98 (0.80–1.19)	0.82
Donor age	1.00 (0.98–1.02)	0.99		
Donor male sex	1.10 (0.99–1.24)	0.08		
Donor race				
- White	1 (ref)	-		
- African American	0.93 (0.82–1.06)	0.29		
- Hispanic	1.07 (0.89–1.28)	0.48		
- Other	0.98 (0.72–1.36)	0.92		
History of hypertension in donor	0.95 (0.84–1.09)	0.46		
KDPI				
- Living donor	0.80 (0.67–0.95)	0.01		
- KDPI < 85	1 (ref)	-		
- KDPI ≥ 85	0.86 (0.68–1.09)	0.20		
HLA A mismatch				
- 0	1 (ref)	-		
- 1	1.37 (1.08–1.73)	0.01		
- 2	1.29 (1.02–1.63)	0.03		
HLA B mismatch				
- 0	1 (ref)	-		
- 1	1.23 (0.91–1.67)	0.19		
- 2	1.24 (0.92–1.66)	0.16		
HLA DR mismatch				
- 0	1 (ref)	-		
- 1	1.25 (1.02–1.53)	0.03		
- 2	1.49 (1.22–1.82)	<0.001		
HLA ABDR mismatch				
- 0	1 (ref)	-	1 (ref)	-
- 1	3.05 (1.11–8.37)	0.03	2.84 (1.03–7.81)	0.04
- 2	4.24 (1.82–9.87)	0.001	4.43 (1.90–10.34)	<0.001
- 3	3.88 (1.71–8.83)	0.001	4.37 (1.92–9.97)	<0.001
- 4	3.69 (1.64–8.33)	0.002	4.32 (1.91–9.80)	<0.001
- 5	4.14 (1.84–9.31)	0.001	5.10 (2.25–11.54)	<0.001
- 6	4.42 (1.95–9.98)	<0.001	5.63 (2.48–12.80)	<0.001
Cold ischemia time	1.00 (0.99–1.01)	0.42		
Allocation type				
- Local	1 (ref)	-		
- Regional	0.86 (0.72–1.03)	0.10		
- National	1.19 (1.02–1.38)	0.03		
Kidney on pump	1.00 (0.89–1.12)	0.99		
Delay graft function	1.44 (1.28–1.61)	<0.001	1.44 (1.28–1.62)	<0.001
EBV status				
- Low risk	0.69 (0.28–1.69)	0.41		
- Moderate risk	1 (ref)	-		
- High risk	1.08 (0.86–1.36)	0.49		
CMV status				
- D−/R−	0.97 (0.80–1.16)	0.72		
- D−/R+	0.94 (0.82–1.07)	0.32		
- D+/R+	1 (ref)	-		
- D+/R−	0.89 (0.75–1.06)	0.20		
Induction immunosuppression				
- Thymoglobulin	0.87 (0.78–0.98)	0.02		
- Alemtuzumab	0.86 (0.74–1.00)	0.05	0.80 (0.68–0.95)	0.01
- Basiliximab	1.47 (1.28–1.68)	<0.001	0.79 (0.64–0.98)	0.03
- Other	0.45 (0.23–0.87)	0.01	1.40 (1.17–1.67)	<0.001
- No induction	1.07 (0.86–1.32)	0.55		
Maintenance Immunosuppression				
- Tacrolimus	0.91 (0.75–1.10)	0.33		
- Cyclosporine	1.86 (1.15–3.00)	0.01	2.33 (1.42–3.82)	0.002
- Mycophenolate	1.35 (1.07–1.70)	0.01		0.02
- Azathioprine	2.26 (1.08–4.75)	0.03	2.70 (1.24–5.87)	0.02
- mTOR inhibitors	2.39 (1.13–5.03)	0.02	2.65 (1.24–5.66)	
- Steroid	1.03 (0.91–1.17)	0.60		

Abbreviations: BMI—Body mass index; CMV—Cytomegalovirus; DGF—Delayed graft function; DM—Diabetes mellitus; EBV—Epstein-Barr Virus; ESKD—End-stage kidney disease; HBs—Hepatitis B surface; HCV—Hepatitis C virus; HIV—Human immunodeficiency virus; HLA—Human leukocyte antigen; KDPI—Kidney donor profile index; mTORi—Mammalian target of rapamycin inhibitors; PRA—Panel reactive antibody; PVD—Peripheral vascular disease.

**Table 3 medicines-08-00066-t003:** Measures of importance for all variables in the forest (The order of these listed variables based on mean decrease in Gini).

Variable	Mean Minimal Depth	Number of Nodes	Accuracy Decrease	Gini Decrease	Number of Trees	Times_a_Root *	*p* Value
BMI	3.1920	72,508	0.0010	236.4036	500	13	<0.001
Cold ischemia time	3.1200	73,014	0.0040	236.2731	500	15	<0.001
Age	2.2080	65,930	0.0036	203.8359	500	64	<0.001
Donor age	3.4700	67,340	0.0022	202.5135	500	5	<0.001
Serum albumin	3.3020	61,652	0.0005	176.4404	500	14	<0.001
PRA	3.2120	40,174	0.0017	113.1232	500	38	<0.001
Education level	3.7800	32,605	0.0001	82.5028	500	20	<0.001
CMV status	4.3320	32,808	0.0002	82.0938	500	1	<0.001
ABO blood type	4.6260	31,967	−0.0000	78.2775	500	2	<0.001
Total HLA mismatch	3.9760	33,012	0.0017	77.9731	500	9	<0.001
Cause of ESKD	3.3980	28,985	0.0019	73.6816	500	43	<0.001
Dialysis duration	4.2220	21,608	0.0008	52.6351	500	11	<0.001
HLA DR mismatch	4.2940	20,334	0.0009	45.7800	500	16	<0.001
HLA A mismatch	5.0020	19,639	0.0003	43.1578	500	4	<0.001
Allocation type	5.0260	16,622	0.0011	38.9588	500	7	1.00
HLA B mismatch	5.3220	16,180	0.0008	37.2973	500	0	1.00
Functional status	5.9100	17,307	0.0000	36.0106	500	0	1.00
Donor male	5.7640	16,565	−0.0000	33.6225	500	0	1.00
Kidney pump use	6.1100	15,487	0.0008	31.9330	500	0	1.00
Steroid use	5.9380	14,707	0.0004	31.7875	500	0	1.00
Male	5.7760	15,622	0.0004	31.6972	500	0	1.00
Working for income	5.9480	13,230	0.0004	28.1003	500	0	1.00
Donor White	6.1580	13,616	0.0003	27.706	500	0	1.00
Hypertensive donor	6.3700	12,578	0.0006	26.6712	500	0	1.00
Thymoglobulin induction	5.7580	12,740	0.0004	26.2509	500	2	1.00
DGF	3.7540	10,816	0.0013	25.4752	500	40	1.00
Donor status	5.4620	11,189	0.0019	>25.2774	500	4	1.00
Donor black	6.3460	11,570	0.0004	24.1010	500	0	1.00
DM	5.7140	11,370	0.0012	23.5846	500	18	1.00
KDPI group	5.5320	9255	0.0020	21.5138	500	7	1.00
Public insurance	6.2720	9871	0.0001	21.2143	500	1	1.00
Basiliximab induction	3.7560	7713	0.0008	20.5859	500	39	1.00
Tacrolimus use	6.4760	7983	0.0002	19.4706	500	0	1.00
Alemtuzumab induction	6.6840	8913	0.0005	19.2899	500	2	1.00
EBV status	6.2540	7302	−0.0000	19.1136	500	0	1.00
Recipient HCV serostatus	6.3560	7444	0.0003	18.7719	500	1	1.00
Donor Hispanic	6.5880	8153	0.0000	18.6623	500	1	1.00
PVD	6.8200	7303	0.0001	17.0666	500	0	1.00
Malignancy	6.9220	6275	0.0001	15.7198	500	1	1.00
Number of transplants	4.3420	4851	0.0019	15.5554	500	38	1.00
No induction	7.1900	6050	−0.0000	14.4486	500	0	1.00
Recipient HIV serostatus	6.0180	4810	0.0003	13.8118	500	6	1.00
MMF use	6.5580	5683	0.0002	13.6395	500	2	1.00
Donor- other race	7.7300	3861	−0.0000	10.5321	500	0	1.00
Preemptive transplant	7.3320	4415	0.0001	9.7186	500	4	1.00
Retransplant	5.6520	3136	0.0019	9.2727	466	61	1.00
Cyclosporine use	6.5829	2157	0.0001	7.7893	499	4	1.00
Recipient HBs antigen	8.1428	2137	−0.0000	6.6728	497	0	1.00
mTORi use	7.5623	1253	0.0000	4.8298	489	4	1.00
Azathioprine use	7.6357	1154	0.0000	4.6491	479	2	1.00
Other induction	10.9853	1263	0.0000	3.2797	471	0	1.00
US resident	12.5317	484	−0.0000	1.5589	328	1	1.00

* Times_a_root—total number of trees in which X_j_ is used for splitting the root node. *p* value < 0.01 means significant variable (the variable is used for splitting more often than would be the case if the selection was random). Abbreviations: BMI—Body mass index; CMV—Cytomegalovirus; DGF—Delayed graft function; DM—Diabetes mellitus; EBV—Epstein-Barr Virus; ESKD—End-stage kidney disease; HBs- Hepatitis B surface; HCV—Hepatitis C virus; HIV—Human immunodeficiency virus; HLA—Human leukocyte antigen; KDPI—Kidney donor profile index; mTORi—Mammalian target of rapamycin inhibitors; PRA—Panel reactive antibody; PVD—Peripheral vascular disease.

**Table 4 medicines-08-00066-t004:** Top 30 interactions that appeared most frequently.

Variable	Root Variable	Mean Minimal Depth	Occurrences	Interaction	Unconditional Mean Minimal Depth
Recipient Age	Age	2.4218	467	Age:Age	2.2080
Donor age	Age	2.4069	467	Age:Donor age	3.4700
Serum albumin	Age	2.6582	461	Age:Serum albumin	3.3020
BMI	Age	2.7236	460	Age:BMI	3.1920
Cold ischemia time	Age	2.5770	459	Age:Cold ischemia time	3.1200
CMV status	Age	3.9216	452	Age:CMV status	4.3320
ABO	Age	4.2978	449	Age:ABO	4.6260
Education level	Age	3.8707	448	Age:Education level	3.7800
PRA	Age	3.6716	448	Age:PRA	3.2120
Total HLA mismatch	Age	3.9931	446	Age:Total HLA mismatch	3.9760
Cause of ESKD	Age	4.4647	442	Age:Cause of ESKD	3.3980
HLA-A mismatch	Age	5.4840	433	Age:HLA-A mismatch	5.0020
HLA-DR mismatch	Age	5.3943	431	Age:HLA-DR mismatch	4.2940
Dialysis duration	Age	5.2511	429	Age:Dialysis duration	4.2220
Functional status	Age	6.5738	426	Age:Functional status	5.9100
HLA-B mismatch	Age	5.9078	426	Age:HLA-B mismatch	5.3220
Steroid use	Age	6.4615	425	Age:Steroid use	5.9380
Working for income	Age	6.6328	425	Age:Working for income	5.9480
Donor male	Age	6.6333	421	Age:Donor male	5.7640
BMI	Cold ischemia time	4.0932	418	Cold ischemia time:BMI	3.1920
Age	Cold ischemia time	4.0842	417	Cold ischemia time:Age	2.2080
Serum albumin	Cold ischemia time	4.2897	417	Cold ischemia time:Serum albumin	3.3020
Kidney pump use	Age	7.2689	416	Age:Kidney pump use	6.1100
Allocation.type	Age	6.4040	414	Age:Allocation type	5.0260
Male	Age	7.0507	414	Age:Male	5.7760
Donor age	Cold ischemia time	4.5872	412	Cold ischemia time:Donor age	3.4700
Donor White	Age	7.3541	411	Age:Donor White	6.1580
BMI	PRA	4.1529	411	PRA:BMI	3.1920
Cold ischemia time	Cold ischemia time	4.3130	407	Cold ischemia time:Cold ischemia time	3.1200
Public insurance	Age	7.6620	406	Age:Public insurance	6.2720

Abbreviations: Age—Recipient age; BMI—Body mass index; CMV—Cytomegalovirus; ESKD—End-stage kidney disease; HLA—Human leukocyte antigen; PRA—Panel reactive antibody.

## Data Availability

Data is available upon reasonable request to corresponding author.

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
