# Peer review of "Feature Importance of Acute Rejection among Black Kidney Transplant Recipients by Utilizing Random Forest Analysis: An Analysis of the UNOS Database"

_medicines, 2021, doi:10.3390/medicines8110066_

Round 1
Reviewer 1 Report
The article is interesting but I think that the results that have been obtained are similar to the general population with kidney transplantation. the comparison with the general population would have been of greater interest.
Author Response
Response to Reviewer#1
Comment #1
The article is interesting but I think that the results that have been obtained are similar to the general population with kidney transplantation. the comparison with the general population would have been of greater interest.
Response: We thank you for reviewing our manuscript and for your critical evaluation. We appreciate the reviewer’s comment. Given Black kidney transplant recipients have had worse allograft outcomes than White recipients, and beneficial effects of immunosuppression with regard to acute rejection are less in Black recipients than those demonstrated in White recipients. In addition, acute rejection in Black kidney transplant recipients are more likely to be steroid resistant. Thus, we conducted this study to assess risk factors and feature importance of acute rejection among Black kidney transplant recipients by utilizing random forest vs. traditional multivariable logistic regression analysis.
We agree with the reviewer that future studies of overall general kidney transplant population patients would also be of interest, and we thus added this important point in the discussion of our study as suggested.
“In addition, future studies assessing tree-based RF feature importance and feature interaction network analysis framework for acute rejection among overall kidney transplant recipient populations are needed.”
Thank you for your time and consideration. We greatly appreciated the reviewer’s and editor’s time and comments to improve our manuscript. The manuscript has been improved considerably by the suggested revisions.

Reviewer 2 Report
The article has the conditions for its publication, I agree with its approval.
Author Response
Response to Reviewer#2
Comment
The article has the conditions for its publication, I agree with its approval.
Response: We thank you for reviewing our manuscript and for your comments. We agree with the reviewer and hope that our novel application of tree-based RF feature importance and feature interaction network analysis framework will be great initiative to provide an understanding of random forest structures to design strategies to prevent acute rejection among Black recipients.
Thank you for your time and consideration. We greatly appreciated the reviewer’s and editor’s time and comments to improve our manuscript. The manuscript has been improved considerably by the suggested revisions.
